# Insights into Canine Infertility: Apoptosis in Chronic Asymptomatic Orchitis

**DOI:** 10.3390/ijms24076083

**Published:** 2023-03-23

**Authors:** Judith Morawietz, Hanna Körber, Eva-Maria Packeiser, Andreas Beineke, Sandra Goericke-Pesch

**Affiliations:** 1Reproductive Unit—Clinic for Small Animals, University of Veterinary Medicine Hannover, Foundation, 30559 Hannover, Germany; 2Department of Pathology, University of Veterinary Medicine Hannover, Foundation, 30559 Hannover, Germany

**Keywords:** autoimmune orchitis, dog, TUNEL, Bcl-2, Bax, Caspase 3, Fas, FasLigand

## Abstract

Chronic asymptomatic orchitis (CAO) is a common cause of acquired non-obstructive azoospermia in dogs. To understand the impact and mode of action of apoptosis, we investigated TUNEL, Bax, Bcl-2, Fas/Fa*s* ligand, and caspase 3/8/9 in testicular biopsies of CAO-affected dogs and compared the results to undisturbed spermatogenesis in healthy males (CG). TUNEL^+^ cells were significantly increased in CAO, correlating with the disturbance of spermatogenesis. *Bcl-2*, *Bax* (*p* < 0.01 each), *caspase 9* (*p* < 0.05), *Fas*, *caspase 8* (*p* < 0.01 each), and *caspase 3* (*p* < 0.05) were significantly increased at the mRNA level, whereas *FasL* expression was downregulated. Cleaved caspase 3 staining was sporadic in CAO but not in CG. Sertoli cells, some peritubular (CAO/CG) and interstitial immune cells (CAO) stained Bcl-2^+^, with significantly more immunopositive cells in both compartments in CAO compared to CG. Bcl-2 and CD20 co-expressing B lymphocytes were encountered interstitially and in CAO occasionally also found intratubally, underlining their contribution to the maintenance of CAO. Our results support the crucial role of the intrinsic and extrinsic apoptotic pathways in the pathophysiology of canine CAO. Autoprotective Bcl-2 expression in Sertoli cells and B lymphocytes seems to be functional, however, thereby also maintaining and promoting the disease by immune cell activation.

## 1. Introduction

Non-obstructive azoospermia (NOA) is a common cause of infertility in breeding male dogs [1,2,3,4]. As affected dogs are often presented at an advanced stage of the disease, the prognosis is usually poor, especially without knowledge and understanding of the cause and degree of disruption of spermatogenesis. Testicular biopsies for histological assessment of the parenchyma are a suitable tool to better elucidate the possible causes of NOA and its consequences, also in dogs [5,6,7]. Recently, we showed that chronic asymptomatic orchitis (CAO) plays an important role in dogs with NOA [5], as indicated by significant lymphoplasmacytic infiltration [7]. Our results are in good agreement with earlier studies mainly based on individual to few cases of canine infertility frequently describing immune cell infiltration, with authors, however, postulating autoimmune orchitis (AIO) [6,8,9,10,11,12,13]. In our recent study on spontaneous infertility in male dogs, the presence of T lymphocytes, macrophages, B lymphocytes, and early and late plasma cells was confirmed [7]. We postulated an important role for T lymphocytes for the development of canine CAO and for B lymphocytes for the progression of the disease.

Whereas CAO/AIO is still poorly characterized in dogs, spontaneous AIO has also been described in black minks [14,15] and rats [16] and is discussed controversially in men [17,18,19] as a cause for acquired infertility. Moreover, experimentally induced orchitis (EAO) in rats and mice is used as a model for chronic testicular inflammation. As it mimics pathological changes found in some human biopsies with impaired spermatogenesis, EAO aims to gain further insights into primary autoimmune orchitis in men [20,21,22] and also chronic testicular inflammation [23,24]. Nevertheless, canine CAO is currently considered idiopathic as the definite cause of the etiology is still missing. Due to the striking similarity of canine CAO to chronic asymptomatic orchitis in men, however, we postulate that gaining deeper insights into canine CAO is also helpful for the understanding of this human infertility cause [5] and might be suitable to develop specific therapeutic options to prevent its development, maintenance, and progression.

Apoptosis, programmed cell death, plays an important role in physiological spermatogenesis [25,26], mainly during puberty [27,28] and also, even though less, during adulthood [26]. Whereas apoptosis is considered important for the removal of abnormal cells and for the maintenance of tissue homeostasis and optimum Sertoli-to-germ-cell ratio under physiological conditions [27,29,30,31], specific triggers, such as toxins, heat, irradiation-inducing DNA damage, or even chromosomal abnormalities, hormonal withdrawal, or pathological conditions such as cryptorchidism [32], can induce also apoptosis [28,31]. Of note, apoptosis also plays an important role in spontaneous and experimentally induced autoimmune orchitis (AIO and EAO) [33,34,35].

Initiation of apoptosis comprises two distinct pathways: the extrinsic or death receptor pathway and the intrinsic or mitochondrial pathway of apoptosis [36,37,38,39], both resulting in the activation of the key effector enzyme of apoptosis, caspase 3, and finally cell death. The extrinsic pathway is activated by an external stimulus, resulting in the activation of the death receptor Fas, allowing for subsequent binding of its ligand (FasLigand, *FasL*) and triggering initiator caspase 8 via the Fas-associated death domain (FADD) [37,40,41,42]. Different from this, the intrinsic, mitochondrial pathway is triggered by intracellular stress signals, such as damaged DNA or metabolic stress. Bax, a proapoptotic cytosolic molecule, and Bcl-2, an antiapoptotic protein of the Bcl-2 family located at the outer mitochondrial membrane (OMM) [43,44,45], act as important counterplayers in the intrinsic pathway [46]. Following a conformational change subsequently to a stressor, Bax migration into the OMM causes increased permeability and cytochrome C release. Cytochrome C together with Apaf-1 and procaspase 9 forms the apoptosome complex activating initiator caspase 9 [36,46,47].

In terms of EAO, Theas et al. [33] identified a significant positive correlation between the overall number of apoptotic and Fas/FasL-expressing cells and the severity of the disease, indicating that apoptosis is involved in the maintenance and progression. Moreover, interstitial lymphocytes, namely T cells, stained immunopositive for FasL, and it was hypothesized that soluble, locally produced FasL can reach the adluminal compartment of the seminiferous tubules in AIO by crossing the cytokine-damaged blood–testis barrier and thereby initiating cell death of germ cells expressing Fas [35]. The role of the death receptor pathway in EAO is further underlined by an increase in procaspase 8, its activated form caspase 8, and caspase 9 [34]. However, factors of the external and internal apoptotic pathways, such as Bcl-2 and Bax, are upregulated in EAO rats ([34]), and a chronology of expression of apoptotic factors was identified in EAO in mice [48]. Whereas initially during increased germ cell apoptosis Bax and Fas expressions are increased due to active inflammation, increased Bcl-2 and FasL characterize the “post-active stage” (resolution of the lymphocytic inflammation) [48].

Up to now, studies on apoptosis in canine patients suffering from infertility due to CAO are lacking. Characterizing the role of apoptosis in canine CAO, however, could significantly contribute to the understanding of the development and progression of the disease. This is of specific interest as CAO in dogs is not induced but spontaneously occurring and resembles the findings in men with AIO/CAO. Consequently, the aim of this study was to investigate apoptosis in testicular tissues of CAO-affected dogs and to compare the results to normal unaffected canine spermatogenesis. Our findings support the crucial role of apoptosis in canine CAO and promote the dog as a possible model system for CAO in men. Moreover, Bcl-2 is a key factor as it enhances Sertoli cell survival but is also important for the “resistance” of B lymphocytes towards apoptosis, thereby contributing to the maintenance and progression of the disease.

## 2. Results

### 2.1. Number of Apoptotic Cells by TUNEL Method

#### Apoptosis Correlates with the Degree of Spermatogenesis Disturbance

The TUNEL method identified only few morphologically apoptotic cells in CG. They were all located at the basal membrane in the tubular compartment and therefore considered to be of germ cell origin (Figure 1). In the case of Sertoli cell-only tubules in CAO, no TUNEL immunopositive (TUNEL^+^) cells were found within the tubule. In CAO samples, morphologically apoptotic TUNEL^+^ cells were found in both compartments, with, however, the majority in the tubular, not in the interstitial compartment (unpaired *t*-test: *p* = 0.0022). Overall, significantly more morphologically apoptotic TUNEL^+^ cells were identified in CAO compared to CG per visual field (unpaired *t*-test: *p* ≤ 0.0001, Figure 1) and also in the tubular compartment (unpaired *t*-test: *p* ≤ 0.0001).

The number of morphologically apoptotic TUNEL^+^ cells was positively correlated with the degree of spermatogenesis disturbance. Despite this, significantly more TUNEL^+^ cells per visual field and within the tubular compartment were found in early arrest (“early”) and late arrest (“late”) compared to CG (visual fields, ANOVA: *p* ≤ 0.0001; Tukey’s test: early—CG *p* ≤ 0.0001, late—CG *p* ≤ 0.0001; tubular compartment, ANOVA: *p* ≤ 0.0001; Tukey’s test: early—CG: *p* = 0.0044, late—CG: *p* ≤ 0.0001).

### 2.2. Intrinsic Pathway of Apoptosis

#### 2.2.1. Bcl-2 Is Upregulated in CAO

RT-qPCR revealed negative results for *Bcl-2* in one dog, which is why only 11 CAO samples were included in the statistical analysis. The ratio (mRNA expression) for *Bcl-2* was significantly higher in CAO compared to CG (unpaired *t*-test: *p* = 0.0092, Figure 2a). ANOVA revealed a significant difference when comparing the groups of early and late arrest and CG (ANOVA *p* = 0.0464) with Tukey’s test, only revealing a trend between early arrest vs. CG (Tukey’s test: early—CG: *p* = 0.0573; late—CG: *p* = 0.3264, Figure 2b).

Western blot revealed a specific protein band at about 25 kDa in the canine testicular protein of a dog with normal spermatogenesis and in Jurkat cells (positive control), indicating the specificity of the antibody (Figure 3). In the negative and isotype controls, no specific band was seen (Figure 3).

Bcl-2 immunopositive (Bcl-2^+^) staining was visible in the cytoplasm of Sertoli cells and some peritubular cells in all samples (Figure 4). Whereas the staining in the tubular compartment was homogenous and light-colored in CG, the intratubular immunopositive signal in CAO was diffuse and strong. This subjective finding was confirmed statistically when comparing the staining intensity in the tubular compartment of CAO vs. CG (Mann–Whitney test: *p* ≤ 0.0001) and also early and late arrest vs. CG (Kruskal–Wallis test: *p* ≤ 0.0001, Dunn’s test: early—CG: *p* = 0.0033, late—CG: *p* = 0.0023). Whereas Leydig cells were not stained in any of the samples, individual or cluster-like Bcl-2^+^ staining was observed in the interstitial compartment of CAO but not CG samples (CAO vs. CG: Mann–Whitney test: *p* ≤ 0.0001). Bcl-2^+^ staining in the interstitial compartment differed also significantly when comparing early and late arrest of spermatogenesis (CAO) to CG (ANOVA: *p* ≤ 0.0001, Tukey’s: early—CG: *p* ≤ 0.0001; late—CG: *p* ≤ 0.0001), with, however, both arrests not differing from each other. Interstitially located Bcl-2^+^ cells were postulated to be immune cells.

#### 2.2.2. Bcl-2 Upregulation in CAO Is Associated with CD20-Expressing B Lymphocytes

To confirm that interstitial Bcl-2^+^ staining in CAO was associated with the presence of immune cells, namely B lymphocytes, we stained serial sections with Bcl-2 and CD20, a well-known specific marker for B cells [49,50]. Whereas no CD20^+^ cells were visible in CG, strong CD20^+^ signals were found in CAO (Mann–Whitney test, *p* ≤ 0.0001), especially in the interstitial compartment. Similarly, CD20^+^ signals differed significantly between early and late arrest vs. CG but not between each other (Kruskal–Wallis test: *p* ≤ 0.0001, Dunn’s test: early—CG: *p* = 0.0007, late—CG: *p* = 0.0034). Interstitial CD20^+^ cells were sometimes solitary but, in several cases, also represented as clusters (Figure 5). Occasionally, CD20^+^ cells were seen within the tubules close to the basal membrane (Figure 5). Few peritubular cells stained CD20^+^ too. In general, CD20^+^ staining was colocalized with Bcl-2^+^ staining as identified in serial sections.

#### 2.2.3. *Bax* and *Caspase 9* mRNA Expression Is Increased in CAO

RT-qPCR revealed negative results for *Bax and caspase 9* in one dog; therefore, only 11 CAO samples were included in the statistical analysis. *Bax* and *caspase 9* ratios (mRNA expression) were significantly higher in CAO compared to CG (unpaired *t*-test: *Bax*: *p* = 0.0023, Figure 6a, *caspase 9*: *p* = 0.013, Figure 7a). Comparing early and late arrest to CG, ANOVA revealed significant differences (*Bax* ANOVA: *p* = 0.0105, *caspase 9* ANOVA: *p* = 0.0334). Whereas *Bax* expression differed significantly from CG in early (Tukey’s test: *p* = 0.0135) and late arrest (Tukey’s test: *p* = 0.0334) (Figure 6b), in the case of *caspase 9*, only late arrest differed significantly from CG (Tukey’s test: *p* = 0.032, Figure 7b).

### 2.3. Extrinsic Pathway

#### *Fas*, *FasLigand*, and *Caspase 8* mRNA Expression

*Fas*, *FasL*, and *caspase 8* mRNA expressions were negative in one dog from CAO, which is why statistical analysis was performed with data of 11 CAO dogs only. Considering the remaining samples, *Fas* and *caspase 8* mRNA expressions (ratio) were significantly higher in CAO compared to CG. Different from this, *FasL* expression was lower in CAO compared to CG, although not significantly (unpaired *t*-test: *Fas*: *p* = 0.0010, Figure 8a; *FasL*: not significant, *p* = 0.1069, Figure 9a; *caspase 8*: *p* = 0.0084, Figure 10a). ANOVA revealed significant differences when comparing ratios of early and late arrest and CG for *Fas* (*p* = 0.0016), *FasL* (*p* = 0.0076), and *caspase 8* (*p* = 0.0058), with higher expressions in early arrest compared to CG for *Fas* (Tukey: *p* = 0.0011, Figure 8b) and *caspase 8* (Tukey: *p* = 0.0044, Figure 10b) but reduced expression for *FasL* (Tukey: *p* = 0.0114, Figure 9b). Results did not differ between late arrest and CG, as well as early and late arrest.

### 2.4. Caspase 3 mRNA and Protein Expression

The *caspase 3* ratio (mRNA expression) was significantly higher in CAO compared to CG (unpaired *t*-test: *p* = 0.0411, Figure 11a). No significant differences were identified when comparing early and late arrest to CG (ANOVA *p* = 0.1090, Figure 11b).

Immunohistochemistry revealed only sporadic immunopositive signals for cleaved *caspase 3* in CAO but not in CG. Immunopositive signals in CAO were predominantly found within the seminiferous tubules (Figure 12).

## 3. Discussion

Although apoptosis was studied in detail under physiological and pathological conditions in the testes of various species, such as humans [51], rats [26,29], and mice [30], and its relevance for the maintenance and progression of EAO in rats [33,34] is well described, only little is known about apoptosis in the canine testis [52,53,54]. Furthermore, its role in “spontaneous” CAO in dogs has not been studied yet, and our study provides first detailed insights in a limited but representative number of samples.

Apoptosis is characterized by DNA fragmentation [36,55], and the TUNEL assay is suitable to detect DNA fragments indicating apoptotic cells [56,57], also in dogs [52,53,58]. Despite the relative ease to perform the assay, it is well known that results have to be interpreted carefully in terms of apoptotic cell demise as early and also late stages of apoptosis can be reversed [59]. Nevertheless, our study is the first to compare morphologically apoptotic TUNEL^+^ cells in CAO-affected and physiological canine spermatogenesis (CG). Different from Kawakami et al. [53], but similar to Henning et al. [52], we identified only few TUNEL^+^ cells in testicular tissues of healthy dogs with semen parameters within the reference range and histologically normal spermatogenesis. The difference might be related to the higher number of healthy animals included in our study (*n* = 10) compared to both previous investigations (*n* = 3). Nevertheless, our results support that DNA strand breakage possibly indicating apoptosis is of minor relevance in the case of unaltered spermatogenesis in adult dogs. Different from this, but similar to EAO in rats [60], the number of morphologically apoptotic TUNEL^+^ cells was significantly increased in CAO. Interestingly, the absolute number of TUNEL^+^ cells was even higher in CAO-affected testis (range: 10–47/30 tubules, *n* = 12) than after repeated scrotal hyperthermia (range: 8.3–12.7/50 tubules, *n* = 3 [52]), emphasizing the relevance of apoptosis in canine CAO. Whereas TUNEL^+^ cells in CG were morphologically apoptotic germ cells only (as described earlier [52]), the increase in morphologically apoptotic cells in CAO is associated with additional TUNEL^+^ staining in the interstitial compartment, also indicating apoptosis in Leydig cells. Sporadic TUNEL^+^ Leydig cells had been reported earlier in three of four azoospermic dogs with increased estradiol-17ß concentrations considered causative [53] and also in other species or cell lines as a consequence of toxicants, such as lead [61], cadmium [62], and lipopolysaccharide (LPS) [63]. The suggested mode of LPS action for induction of apoptosis in murine Leydig cell culture is via macrophage activation, thereby also inducing inflammatory changes [63]. An important role of macrophages in the initiation of canine AIO/CAO was postulated earlier [7], indicating possibly a similar mechanism in dogs. Further investigations into macrophage markers and secretion products and into apoptosis in canine CAO testis can support this hypothesis.

To gain deeper insights into the mode of action of apoptosis, we investigated different parameters of the intrinsic (*Bcl-2*, *Bax*, *caspase 9*) and extrinsic pathways (*Fas*, *FasL*, *caspase 8*). Due to the lack of specific canine antibodies and/or lacking cross-reactivity of available antibodies with canine testicular tissues, the expression of involved factors was predominantly studied at the mRNA level, allowing mainly for general conclusions on both pathways in canine CAO but not for details on cellular localizations and their CAO-associated changes. Further studies should include investigations at the post-transcriptional level, e.g., by use of custom-made antibodies to overcome the lack of suitable canine cross-reacting antibodies, to confirm and quantify expression changes in apoptosis-related proteins.

As in EAO in rats [34], *Bax* and *caspase 9* mRNA expressions were significantly increased in canine CAO testis compared to CG. The proapoptotic Bax and the anti-apoptotic Bcl-2 form heterodimers [32,64,65], and the Bax/Bcl-2 ratio is known to be critically involved in the determination of cell fate—survival or death [65,66]. Consequently, upregulation of Bax and also of Bcl-2, as observed in CAO, will cause dysregulation of the respective ratio and consequently result in or prevent apoptosis. Even though Bax expression is well understood during testicular development and for density-dependent regulation of spermatogonia in murine spermatogenesis [67], little to nothing is known in adult, mature testis, and also in dogs. Unfortunately, due to the lack of suitable canine or canine cross-reacting Bax antibodies, we were only able to investigate Bcl-2 protein expression on cellular localization, confirming not only upregulated Bcl-2 mRNA but also protein expression. We identified Bcl-2 protein expression in Sertoli cells of all investigated canine testicular samples, with normal (CG) and disrupted spermatogenesis (CAO). It is well known that cultured Sertoli cells express Bcl-2 in rats [68] and mice [69]. The observation that Sertoli cells are often the only cell type surviving pathological conditions such as EAO in rats and impaired spermatogenesis with focal inflammatory lesions in humans resulted in the hypothesis that Sertoli cells are resistant to apoptosis [70]. Upregulation of Bcl-2 in canine CAO Sertoli cells might indicate a counterregulatory mechanism to increased apoptosis. It remains to be clarified whether Sertoli cells try to counteract apoptotic stimuli in order to protect the (remaining) germ cells or whether Bcl-2 “overexpression” aims for the self-protection of Sertoli cells. However, according to the authors’ clinical experiences, as CAO results in irreversible infertility, it appears that the survival of Sertoli cells can “rescue” only individual germ cells but is not sufficient to maintain spermatogenesis and prevent male infertility. On the other hand, possible overexpression of Bax in spermatogonia might be causative for increased germ cell apoptosis in CAO, explaining why reduction in spermatogonia and spermatogonial stem cells in CAO [71]. The use of custom-made canine anti-Bax antibodies could help in proving this hypothesis.

Beyond expression in Sertoli cells, interstitial immune cells and some peritubular cells but not germ cells, stained Bcl-2^+^. Whereas a supportive role for spermatogenesis might be postulated for Bcl-2 expressing peritubular cells in dogs in the case of CAO, there is no doubt that the significant immune cell infiltration is rather detrimental for spermatogenesis, causing disruption of spermatogenesis (late/early arrest), germ cell depletion up to Sertoli cell-only syndrome, fibrosis, etc., as clearly visible in CAO [5]. Bcl-2 protein expression in immune cells under physiological and pathological conditions is well known in B and T lymphocytes [72,73]. As we previously identified a significant increase in immune cells in canine AIO/CAO and postulated an important role of B lymphocytes for the progression of the disease due to the second humoral immunological response [7], we stained serial sections with Bcl-2 and CD20, a well-known specific marker for B cells [49,50]. Significantly more CD20^+^ cells were identified in CAO compared to CG. In serial sections, co-expression of CD20 and Bcl-2, especially in the interstitial compartment, confirmed Bcl-2^+^ B lymphocytes. As overexpression of Bcl-2 leads to the failure of apoptosis of B lymphocytes in mice [74,75] and also certain human B-cell lymphomas [76,77], we hypothesize that increased Bcl-2 expression in CAO samples prevents immune cell death. It seems likely that upregulation of NF-κB (unpublished data) is involved in the induction of antiapoptotic Bcl-2 and family genes, thereby mediating B-cell survival [78]. These findings support our earlier hypothesis of B lymphocytes being crucial for the maintenance and progression of CAO and also emphasize the importance of Bcl-2. Since few CD20^+^-Bcl2^+^ cells were also present in the seminiferous tubules in CAO, we further hypothesized that B lymphocytic infiltration of the tubular compartment is related to the break-down of the blood–testis barrier in CAO [12,79], likely by secretion of various cytokines [80] and subsequent induction of germ cell apoptosis [81,82,83].

Not only parameters of the intrinsic pathway but also mRNA expressions of *Fas* and *caspase 8* but not *FasL*, all involved in the extrinsic pathway, were significantly overexpressed in CAO testis samples compared to CG. Similarly, we recently identified tumor necrosis factor α (TNF-α) interacting with the TNF-α-receptor 2 (TNFR2) as an important inducer of germ cell apoptosis in canine CAO as well [84]. Activation of both the intrinsic and extrinsic pathways results in the activation of the effector caspase 3 [40], as confirmed by increased *caspase 3* mRNA expression and caspase 3^+^ protein expression in CAO only. Comparably to our results, caspase 3 expression was not identified in undisturbed rat spermatogenesis [33], was almost absent in the normal human testis [85], and was present in altered spermatogenesis as orchitis in rats [33], cryptorchidism in dogs [86], and Sertoli cell-only syndrome and maturation arrest in men [87], supporting its significant role in pathological conditions of the testis.

Interestingly and different from EAO in rats, *FasL* was not significantly increased in canine CAO compared to CG. FasL is important for maintaining the immune privilege of organs [88,89], such as the testis [21,90], by regulating Fas-expressing T cells [88]. The low *FasL* mRNA expression in dogs with CAO might be related to the failure of elimination or even the accumulation of *Fas*-expressing T cells and/or loss of the immune privilege of the testis associated with CAO/AIO. This goes along with the significantly increased numbers of CD3^+^ T cells in canine CAO compared to healthy testis [7]. The present findings support our hypothesis that the pathogenesis of canine CAO is related to (macrophage-induced T) lymphocyte activation with T lymphocytes, inducing a delayed immunological response and resulting in the development and establishment of canine CAO [7]. Further studies should investigate whether Fas/FasL is expressed by canine T cells, as described during the development of EAO in rats [35], or by macrophages. Moreover, studies might prove whether the soluble form of FasL plays a role in chronic canine CAO for the induction of germ cell apoptosis, as previously described in chronic EAO in rats [35]. Last but not least, FasL expression should be studied at the cellular (protein) level, and compartment-related mRNA analysis of laser-assisted cell-picked tubular or interstitial tissue should also be considered.

Based on the present results and the knowledge about the detrimental outcome of canine CAO, further studies should investigate the effect of drugs specifically affecting or modulating apoptosis. Drugs, such as a specific nitric oxide synthase inhibitor (N omega-nitro-L-arginine methyl ester, abbreviated as L-NAME) associated with a reduction in Bax/Bcl-2 ratio and cytochrome c [91], or nortriptyline, a second-generation antidepressant acting as an inhibitor of mitochondrial outer membrane permeability (MOMP) [92] or caspase 9 [93], might prevent the development and progression of CAO in dogs and enhance testicular function. It remains, however, challenging to reduce specifically apoptosis in one cell type, namely germ cells, and to increase it in another, namely immune cells.

## 4. Materials and Methods

### 4.1. Study Design, Animals, and Grouping

The study included testicular samples from 22 sexually mature, clinically healthy male dogs. Twelve dogs were diagnosed with chronic asymptomatic orchitis (CAO) in bilateral testicular biopsies [5,7]. Of those, nine sired successfully at least one litter before presentation due to infertility. Detailed clinical procedures to identify the cause of azoospermia, verify NOA/CAO, and rule out infectious causes had been described earlier. Briefly, clinical and andrological examination, including semen collection and analysis, ultrasound of the genital tract, determination of the alkaline phosphatase (AlP) in the seminal plasma, bacteriological examination of the ejaculate, serological testing for Brucella canis, and endocrine analysis (testosterone/estradiol-17ß/thyroxine/free thyroxine/cTSH/thyroid antibodies), was performed in infertile dogs. Testicular specimens of the remaining ten dogs represented normal spermatogenesis (confirmed by repeated normospermic ejaculates) [94], assigning them to the control group (CG). The mean age of the dogs was 6.0 ± 1.7 years (3.2–9.5 years) in CAO and 3.1 ± 3.2 (0.9–9.9 years) in CG, respectively. Dogs belonged to various breeds, namely Beagle (*n* = 5), Collie (*n* = 3), Cairn Terrier, Miniature Poodle, Labrador Retriever, Icelandic sheepdog, Welsh Corgi Pembroke, Coton de Tulear, Jack Russell Terrier, German shepherd, Havanese, Maltese, Boston Terrier, and Chihuahua; one dog was a mongrel.

Permission for all samples was received by the respective authorities. Collection of CAO samples for diagnostic purposes was approved by the Dyreforsøgstilsynet Fødevarestyrelsen and by the local animal ethics committee of the University of Veterinary Medicine Hannover, Foundation. Testis samples from control animals were taken from dogs that were presented for routine castration for other than health issues. All owners agreed on further use of samples for research purposes.

### 4.2. Study Design, Sample Collection, and Processing

Testicular specimens were obtained from each testis under general anesthesia either by scissors biopsy (CAO) or after castration (CG). In the CAO group, arrest of spermatogenesis was further histologically characterized into early or late arrest of spermatogenesis depending on the arrest of differentiation of germ cells in the majority of round tubules. Samples of five dogs were assigned to the group “early arrest”, with Sertoli cell only or spermatogonia being identifiable. The remaining seven dogs were assigned to the group “late arrest”, with spermatogenesis being arrested at a later level.

Tissue processing, fixation, and embedding was performed as previously described [95]. One part of the testicular sample was stored in RNAlater^TM^ RNAprotect Tissue Reagent (#76106; Qiagen GmbH, Hilden, Germany) at −80 °C until being used for mRNA and protein extraction. Bouin’s fixative solution was used for 24 h for the other part. Following several washing steps with 70% ethanol and dehydration, the samples were embedded in paraffin for histology and immunohistochemistry.

### 4.3. Quantitative Real-Time PCR (RT-qPCR)

Total RNA from RNAlater^TM^-immersed testicular tissue was isolated from samples of each dog according to the manufacturer’s protocol. Only mRNA samples with a ratio of absorbance at 260 nm and 280 nm between 1.95 and 2.05 were considered for further investigation. Full-length first-strand cDNA synthesis was performed by using 200 ng/µL RNA and the RevertAidFirst Strand cDNA Synthesis Kit (#K1622; Thermo Scientific, Waltham, MA, USA) according to the manufacturer’s protocol including DNase treatment. A non-template control (autoclaved water instead of RNA) was included in every run for quality control. To test for the expression of *Bcl-2*, *Bax*, *Fas*, *FasLigand*, and *caspase 3*, *8*, and *9*, primer sets were used (Table 1) using known sequences from GenBank. For RT-qPCR, 3 µL cDNA (dilution 1:10) was added to 8 µL FastStart Essential DNA Green Master (#06402712001; Roche Diagnostics GmbH, Mannheim, Germany), 1 µL forward and reverse primer (10 pmol) (Table 1). and 2 µL sterile Aqua bidest. RT-qPCR conditions were 95 °C for 10 min, followed by 35 cycles of 95 °C for 10 s, 60 °C (*caspase 8*: 62 °C) for 10 s, 72 °C for 10 s, and a melting curve with 65–97 °C. All samples were run in triplicates using a LightCycler^®^96 real-time PCR system (Software Release 1.5.0, Version 1.5.0.39, Roche Diagnostics GmbH); *glyceraldehyde-3-phosphate dehydrogenase* (*GAPDH*) and *beta-actin* (*ß-actin*) served as reference genes. A negative control (above-mentioned water) was included in every RT-qPCR. For calculation of PCR efficiencies of target and reference genes, the Roche LightCycler^®^96 SW 1.5.0.39 software was applied by using a relative standard curve derived from a triplet RT-qPCR run of a twofold dilution series (1:2–1:128) of pooled cDNA samples, whereas the efficiency (E) was E = 10(−1/m), with m being the slope of linear regression line [96]. Table 1 shows the respective efficiencies. For evaluation of the RT-qPCR results, an efficiency-corrected relative quantification according to [96] was performed. As *GAPDH* showed the slightest variation in mRNA expression compared to *ß-actin*, *GAPDH* was chosen as a reference gene for calculations.

### 4.4. Protein Extraction and Western Blot Analyses

Western blotting was performed to confirm the species-reactivity of the Bcl-2 antibody applied; TUNEL [52,97] and cleaved caspase 3 [97] had been confirmed to be specific in canine tissue samples earlier. Tissue processing for protein extraction and determination of protein concentration was described earlier [98]. Approximately 100 µg protein was used for the Western blot following the previously described protocol [98] using, however, the primary Bcl-2 monoclonal antibody (Clone Bcl-2/100/D5, #NCL-L-bcl-2, Novocastra™, Leica Biosystems Newcastle Ltd., Newcastle, UK, protein concentration 4 mg/mL, dilution 1:200) and the respective secondary antibody (BA-2000, Vector Laboratories, Burlingam, CA, USA, dilution 1:500). Visualization was performed by the ChemiDoc Imaging System™ with ImageLab™ Touch Software (ImageLab 6.0.1, Bio-Rad Laboratories, Hercules, CA, USA). Blocking buffer only was used as negative control, and an irrelevant mouse IgG (I-2000, Mouse IgG, Vector Laboratories) and its respective concentration served as isotype control. Jurkat whole-cell lysate (#sc-2204; Santa Cruz Biotechnology, Inc., Dallas, TX, USA) served as positive control based on the manufacturer’s recommendation and the definition of positive controls according to Pillai Kastoori et al. [99]. Based on this, a positive control is a lysate of cell lines or tissues of the species against which the antibody is raised. As this antibody is against human Bcl-2, this justifies the use of the human Jurkat cell line as positive control. No further canine tissues possibly expressing Bcl-2 were included to further confirm cross-reactivity in the dog besides canine testicular tissue.

### 4.5. TUNEL Method, Procedure, and Evaluation

Apoptotic cells were identified by detection of DNA fragments by terminal deoxyribonucleotidyl transferase (TdT)-mediated dUTP nick end-labeling (TUNEL) using the ApopTaq^®^ Peroxidase Detection Kit (#S7100; Chemicon International (Millipore, Temecula, CA, USA) according to the manufacturer’s instructions with minor modifications. Following deparaffinization, the 2 µm thick sections were pretreated in cooking citrate buffer (pH = 6) and then washed twice in distilled water. To block endogenous peroxidase activity, slides were quenched in 3% hydrogen peroxide in methanol for 5 min at room temperature. Followed by two washing steps in PBS (#1112.2; phosphate-buffered saline, pH 7.4, Roti^®^fair, Carl Roth GmbH & Co.KG, Karlsruhe, Germany) for 5 min each, the equilibration buffer was applied directly onto the tissue samples, and slides were incubated for 30 min at room temperature. Right after this step, each sample was incubated in a humidified chamber for 1 h at 37 °C with the reaction buffer containing TdT enzyme (dilution 1:20). The negative control was incubated with PBS instead. To stop the reaction, slides were placed in stop/wash buffer for 10 min and then washed in PBS three times for 1 min each. Anti-digoxigenin conjugate was applied to each sample in a humidified chamber for 30 min at room temperature. This was followed by 4 washing steps with PBS for 2 min each, and then the tissue sections were stained with DAB (#DC137C100; DCS Chromoline, Innovative Diagnostik Systeme) for 2 min. Slides were counterstained with hematoxylin and after dehydration covered with HistoKitt (#41025; Assistant, Osterode, Germany). Immunopositive signals were counted in 20 randomly selected visual fields (10 per side, CAO) using an Olympus BX41TF Microscope (Olympus^®^, Tokyo, Japan) with an Olympus DP72 camera (Olympus Corporation, Tokyo, Japan) and the Olympus cellSense Dimension Software (version 2.1, Olympus Corporation, Tokyo, Japan) at 200-fold magnification and assigned to the tubular or interstitial compartment. Additionally, immunopositive signals in 30 randomly selected tubules (15 per side, CAO) were evaluated. All dark brown signals that could be clearly associated with a cell were considered immunopositive and consequently counted.

### 4.6. Immunohistochemistry and Evaluation of Bcl-2, CD20, and Caspase 3 Staining

Immunohistochemistry (IHC) was performed as previously described [100]. Tissue sections of 2 µm thickness were first deparaffinized and rehydrated followed by antigen demasking by cooking with citrate buffer (pH = 6). Endogenous peroxidase activity was blocked using 3% hydrogen peroxide in methanol. In order to block unspecific antibody binding sites, sections for Bcl-2 IHC were first incubated with 10% horse serum (S-2000, Vector Laboratories, Burlingame, CA, USA) and 3% bovine serum albumin (#0175-100 g; BSA, VWR Life Science, Solon, OH, USA) in ICC buffer (1.2 g Na_2_HPO_4_, 0.2 g KH_2_PO_4_, 0.2 g KCl, 8.0 g NaCl, 3 mL Triton ad 1000 mL), washed with ICC buffer followed by incubation with the primary Bcl-2 monoclonal antibody (Clone bcl-2/100/D5, #NCL-L-bcl-2, Novocastra™, dilution 1:200, corresponding to 0.02 mg/mL) overnight at 4 °C. On the next day and following washing steps with ICC, the slides were first incubated with SuperVision-2 HRP Enhancer (#PD000KIT; DCS, Innovative Diagnostik Systeme, Hamburg, Germany) for 20 min, washed again, and then incubated with HRP Polymer (DCS, Innovative Diagnostik Systeme) for 20 min. Visualization of the signals was performed with DAB (DCS Chromoline, Innovative Diagnostik Systeme) according to the manufacturer’s protocol. For the negative control ICC buffer only was applied and an irrelevant mouse IgG antibody (I-2000, Mouse IgG, Vector Laboratories) in the respective protein concentration served as isotype control. Canine lymph nodes served as positive control. Slides were slightly counterstained with hematoxylin and mounted with HistoKitt. Bcl-2 staining was evaluated in the tubular compartment and the staining intensity was determined in 30 round tubules (15 each side, CAO) by using a grading system according to [101] and differentiating mild (1), moderate (2), and strong (3) staining. Moreover, the percentage of immunopositive cells was semiquantitatively assessed in 20 randomly selected visual fields (10 each side, CAO) in the interstitial compartment and categorized as 0 (0% positive cells), 1 (>0–20%), 2 (20–40%), 3 (40–60%), 4 (>60%) [101].

For CD20 (polyclonal CD20, #PA5-16701, ThermoScientific, dilution 1:300, corresponding to 0.67 µg/mL) immunohistochemistry, the modifications were as follows: Blocking of unspecific binding sites was performed with 10% goat serum (S-1000, Vector Laboratories) in ICC buffer. Subsequent procedures and staining were performed as described above for Bcl-2 using SuperVision-2 HRP Enhancer (DCS, Innovative Diagnostik Systeme), HRP Polymer (DCS, Innovative Diagnostik Systeme) and DAB (DCS Chromoline, Innovative Diagnostik Systeme) according to the manufacturer’s protocol. Negative controls (ICC buffer only) and isotype controls (I-1000, Rabbit IgG, Control Antibody, Vector Laboratories) at corresponding protein concentrations were included. Slides were counterstained using hematoxylin and following dehydration mounted with HistoKitt. For evaluation of CD20, all positive signals, which could be assigned to a cell, were counted in 20 (10 each side, CAO) randomly selected visual fields at 200-fold magnification. Positive signals were classified as in the either tubular or interstitial compartment.

IHC against caspase 3 using a cleaved caspase 3 (Asp175) polyclonal antibody (#9661, Cell Signaling Technology^®^, dilution 1:50, corresponding to 1.04 mg/mL) was performed as described above with, however, the following modifications: 10% goat serum (S-1000, Vector Laboratories) in ICC buffer was used for blocking of unspecific binding sites and BA-1000 (Vector Laboratories) as secondary antibody. Visualization of immunopositive signals was performed by an immunoperoxidase system (#PK-6100; VECTASTAIN Rabbit IgG Elite ABC Kit and #SK-4800; Vector Nova-RED Substrate Kit, Vector Laboratories) according to the manufacturer’s instructions. Samples were run with negative (ICC buffer only) and isotype controls (I-1000, Rabbit IgG, Vector Laboratories) at corresponding protein concentrations. Canine spleen served as positive control. Slides were counterstained with hematoxylin and covered with HistoKitt. Cleaved caspase 3 immunopositive signals were descriptively evaluated at 200-fold and 400-fold magnification. Microscopical evaluation of all IHCs was performed using an Olympus BX41TF Microscope (Olympus^®^) with an Olympus DP72 camera (Olympus Corporation) and the Olympus cellSense Dimension Software (version 2.1, Olympus Corporation) as described above in the given magnification.

### 4.7. Statistical Analysis

GraphPad Prism 7 software (GraphPad Software, Inc., La Jolla, CA, USA) and Microsoft Excel (Microsoft^®^ Excel, Version 16.66.1, Microsoft Corporation, Washington, DC, USA) were applied for all statistical analysis. Values at a level of *p* < 0.05 were considered to be statistically significant.

In the case of mRNA expression (ratio) of *Bcl-2*, *Bax*, *Fas*, FasLigand, and *caspase 3*, *8*, and *9,* the Shapiro–Wilk and Kolmogorov–Smirnov test verified normal distribution, so data were presented as arithmetic mean and standard deviation ($\bar{x}$ ± SD). To test for significant differences between CAO and CG, an unpaired *t*-test was applied. For comparison of early and late arrest to CG, ANOVA was used followed by Tukey’s test, if ANOVA revealed *p* < 0.05.

Cleaved caspase 3 protein expression was analyzed descriptively. TUNEL, Bcl-2, and CD20 protein expression were analyzed statistically. For all these results, individual results obtained from the left and right testes of each dog were tested for significant differences by paired *t*-test. Results did not differ allowing summarizing results per dog. Subsequently, for comparison of CAO and CG, data were analyzed for normal distribution using the Shapiro–Wilk and Kolmogorov–Smirnov tests. For TUNEL results, testing verified the normal distribution of raw data of the visual fields and log-transformed data of the tubular compartment, so an unpaired *t*-test was used to test for significant differences between CAO and CG TUNEL results. Moreover, TUNEL and CD20 immunopositive signals in the tubular and interstitial compartments were compared statistically in CAO by *t*-test. As CD20 results were not normally distributed, the Mann–Whitney test was applied for comparison of CAO and CG. For Bcl-2, neither the results of the staining intensity in the tubular compartment nor the immunopositive cells in the interstitial compartment were normally distributed why the Mann–Whitney test was used for further analysis.

For comparison of early and late arrest and CG, data were again tested for normal distribution as described above. Subsequent tests were chosen based on normal or non-normal data distribution and if overall group comparison revealed *p* < 0.05: For TUNEL, an ANOVA for multiple comparisons was applied and followed by Tukey’s test. Due to the not normal CD20 data distribution, the Kruskal–Wallis test was used followed by Dunn’s multiple comparisons. For Bcl-2 results obtained from the tubular compartment, the Kruskal–Wallis test was applied followed by Dunn’s test, and in the case of the percentage of immunopositive cells in the interstitial area, ANOVA followed by Tukey’s test was used.

Data are presented as arithmetic mean and standard deviation ($\bar{x}$ ± SD) (all ratios, TUNEL visual fields), as geometric mean and dispersion factor (TUNEL tubular compartment), or as median with first and third quartiles (CD20, Bcl-2), including in the range in selected cases. Moreover, the range is given for data obtained from immunohistochemistry.

## 5. Conclusions

The identified upregulation of factors of the intrinsic and extrinsic pathways and of cleaved caspase 3 underlines the important role of apoptosis in the pathophysiology of CAO. Upregulated Bcl-2 expression seems to be involved in Sertoli cell survival but does not prevent germ cell apoptosis and irreversible damage of spermatogenesis. As postulated earlier, B lymphocytes (Bcl-2^+^-CD20^+^) are involved in the maintenance and progression of CAO, *and Bcl-2 might* be the key factor preventing their apoptosis. Given the numerous parallels in the pathogenesis and maintenance of human and canine AIO, our results might as well be beneficial for both species.

## Figures and Tables

**Figure 1 ijms-24-06083-f001:**
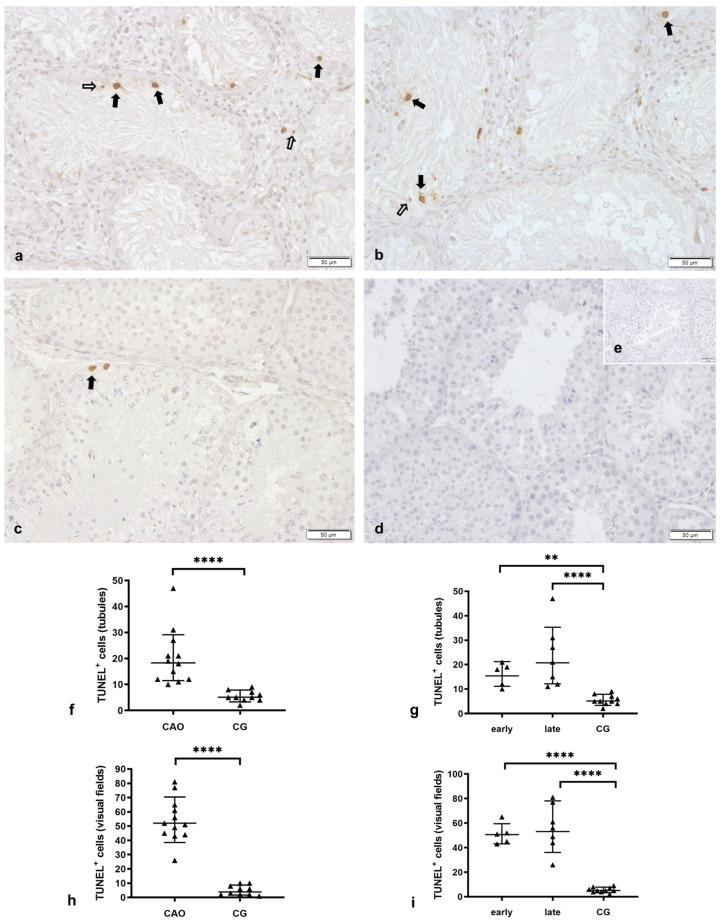
**The number of morphologically apoptotic TUNEL^+^ cells was positively correlated with the degree of spermatogenesis disturbance.** Morphologically apoptotic TUNEL immunopositive (TUNEL^+^) cells in testicular samples (**a**,**b**) from dogs with chronic asymptomatic orchitis (CAO, *n* = 12) and from normospermic, healthy controls (CG; *n* = 10), (**d**) isotype control, (**e**) negative control (given as insert); ((**a**–**e**) 200× magnification) 🠮 (black bold arrow) TUNEL^+^-positive cells, ⇨ (white arrow) apoptotic (TUNEL^+^) cell debris. Number of apoptotic TUNEL^+^ cells per (**f**,**g**) 30 tubules or per (**h**,**i**) 20 visual fields comparing (**f**,**h**) CAO (*n* = 12) and CG (*n* = 10) and (**g**,**i**) CAO samples with early (*n* = 5) and late arrest (*n* = 7) of spermatogenesis (early/late) and normospermic healthy controls (CG, *n* = 10). ** *p* < 0.01; **** *p* < 0.0001.

**Figure 2 ijms-24-06083-f002:**
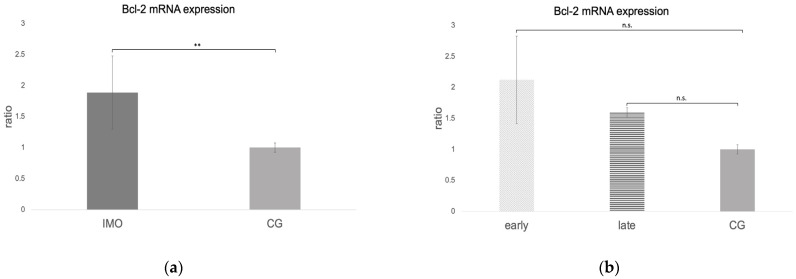
*Bcl-2* mRNA expression (ratio) in testicular samples obtained from dogs suffering from chronic asymptomatic orchitis (CAO) and healthy control dogs (CG). (**a**) Ratios of CAO (*n* = 11) and CG (*n* = 10) and (**b**) ratios of CAO early arrest (*n* = 5) and CAO late arrest (*n* = 6) vs. CG (*n* = 10) are presented. Results are presented as arithmetic mean and standard deviation ($\bar{x}$ ± SD). Datasets with different asterisks differ significantly: ** *p* < 0.01, n.s. = not significant.

**Figure 3 ijms-24-06083-f003:**
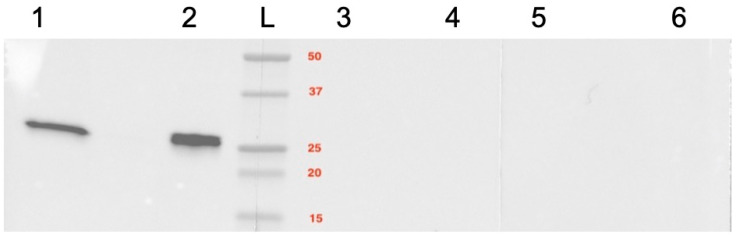
Western Blot analysis for *Bcl-2*. Lanes 1 and 2 were incubated with the specific antibody (positive control); Lanes 3 and 4 were incubated with the corresponding isotype control. In Lanes 5 and 6 the primary antibody was omitted (negative control). Lanes 1, 4 and 5: Jurkat cell lysate; Lanes 2, 3, and 6: canine testis tissue homogenate. Molecular weight markers are expressed in kDa; L = ladder.

**Figure 4 ijms-24-06083-f004:**
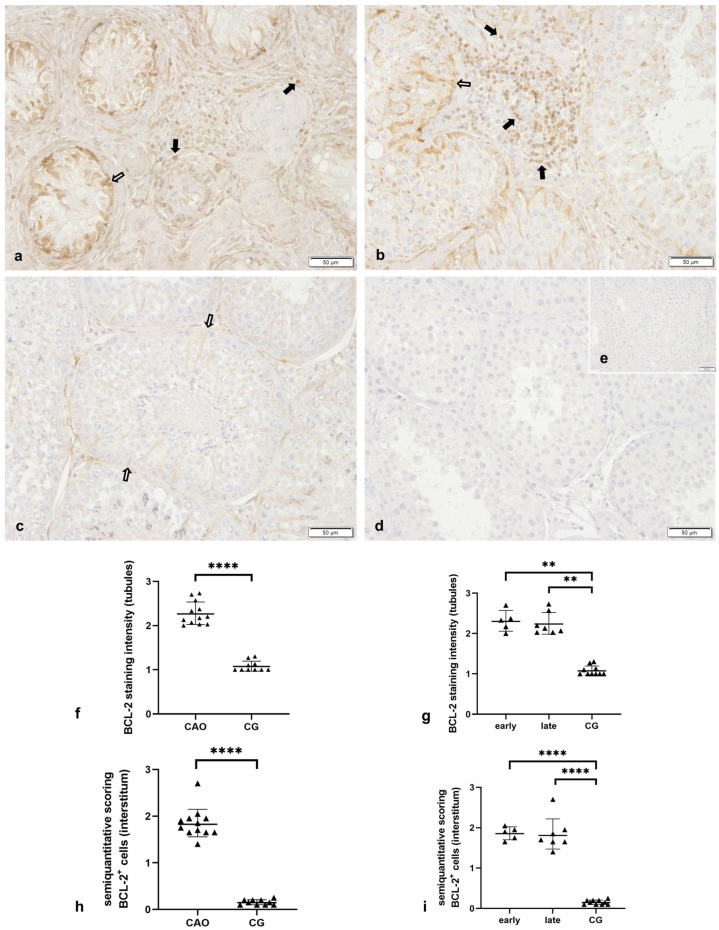
**Numerous interstitially located cells stained Bcl-2^+^ in chronic asymptomatic orchitis (CAO), besides Sertoli cells and some peritubular cells expressed Bcl-2 in normospermic healthy controls (CG) and CAO samples.** Immunostaining for *Bcl-2* in testicular samples of (**a**,**b**) dogs with CAO (*n* = 12) and (**c**) from normospermic healthy controls (CG, *n* = 10), (**d**) isotype controlInd (**e**) negative control (given as insert); ((**a**–**e**) 200× magnification) 🠮 (bold black arrow) *Bcl-2* immunopositive (Bcl-2^+^) immune cells, ⇨ (white arrow) Bcl-2^+^ Sertoli cells. (**f**,**g**) Staining intensity of *Bcl-2* in 30 round tubules representing 1—mild, 2—moderate, and 3—strong staining comparing (**f**) CAO (*n* = 12) and CG (*n* = 10) and (**g**) CAO samples with early (*n* = 5) and late arrest (*n* = 7) of spermatogenesis (early/late) and normospermic healthy controls (CG, *n* = 10). (**h**,**i**) Semiquantitative assessment of the percentage of Bcl-2^+^ cells in the interstitial compartment of 20 visual fields comparing (**h**) CAO (*n* = 12) and CG (*n* = 10) and (**i**) CAO samples with early (*n* = 5) and late arrest (*n* = 7) of spermatogenesis (early/late) and normospermic healthy controls (CG, *n* = 10). The following categories exist: 0: 0%, 1: >0 to 20%, 2: 20 to 40%, 3: 40 to 60%, 4: ≥ 60% Bcl-2^+^ cells. ** *p* < 0.01, **** *p* < 0.0001.

**Figure 5 ijms-24-06083-f005:**
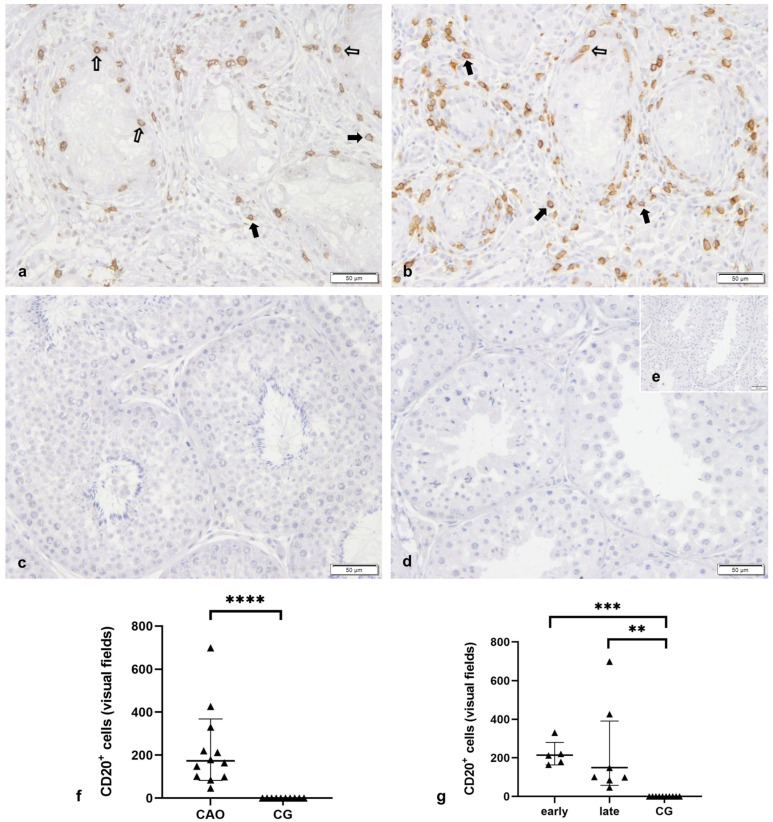
**CD20 staining was absent in normospermic healthy canine testis, but strongly expressed in chronic asymptomatic orchitis (CAO).** Immunostaining for CD20 in testicular samples of (**a**,**b**) dogs with CAO (*n* = 12) and (**c**) from normospermic healthy controls (CG, *n* = 10), (**d**) isotype control and (**e**) negative control (given as insert); ((**a**–**e**) 200× magnification) 🠮 interstitial CD20 immunopositive (CD20^+^) cell, ⇨ tubular CD20^+^ cell (**f**,**g**) Number of CD20^+^ cells per 20 visual fields comparing (**f**) CAO (*n* = 12) and CG (*n* = 10) and (**g**) CAO samples with early (*n* = 5) and late arrest (*n* = 7) of spermatogenesis (early/late) and normospermic healthy controls (CG, *n* = 10). ** *p* < 0.01, *** *p* < 0.001, **** *p* < 0.0001.

**Figure 6 ijms-24-06083-f006:**
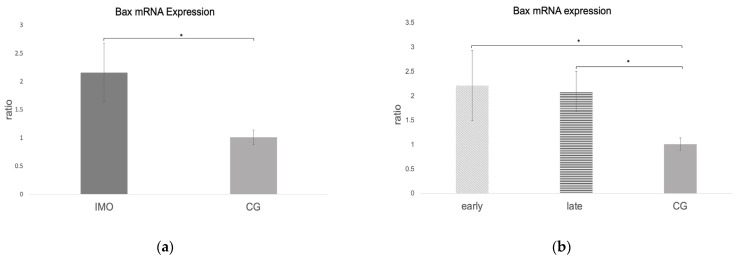
*Bax* mRNA expression (ratio) in testicular samples obtained from dogs suffering from chronic asymptomatic orchitis (CAO) and healthy control dogs (CG). (**a**) Ratios of CAO (*n* = 11) and CG (*n* = 10) and (**b**) ratios of CAO early arrest (*n* = 5) and CAO late arrest (*n* = 6) vs. CG (*n* = 10) are presented. Results are presented as arithmetic mean and standard deviation ($\bar{x}$ ± SD). Datasets with different asterisks differ significantly: * *p* < 0.05.

**Figure 7 ijms-24-06083-f007:**
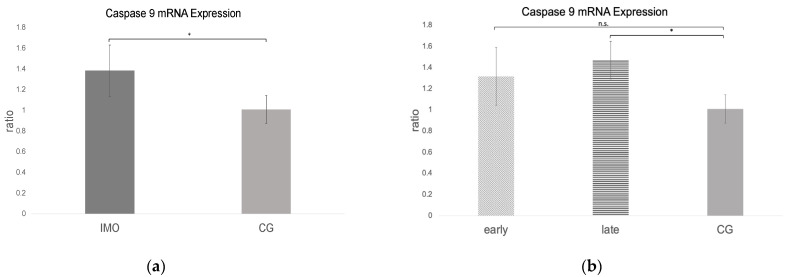
*Caspase 9* mRNA expression (ratio) in testicular samples obtained from dogs suffering from chronic asymptomatic orchitis (CAO) and healthy control dogs (CG). (**a**) Ratios of CAO (*n* = 11) and CG (*n* = 10) and (**b**) ratios of CAO early arrest (*n* = 5) and CAO late arrest (*n* = 6) vs. CG (*n* = 10) are presented. Results are presented as arithmetic mean and standard deviation ($\bar{x}$ ± SD). Datasets with different asterisks differ significantly: * *p* < 0.05; n.s. = not significant.

**Figure 8 ijms-24-06083-f008:**
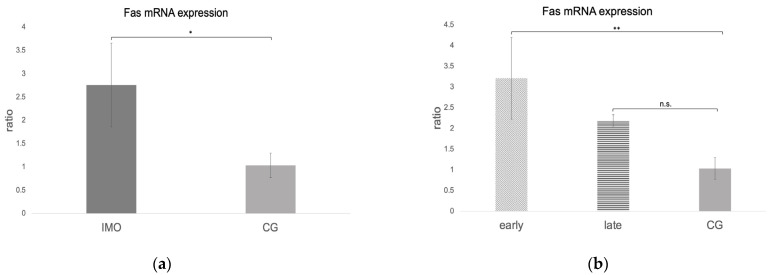
*Fas* mRNA expression (ratio) in testicular samples obtained from dogs suffering from chronic asymptomatic orchitis (CAO) and healthy control dogs (CG). (**a**) Ratios of CAO (*n* = 11) and CG (*n* = 10) and (**b**) ratios of CAO early arrest (*n* = 5) and CAO late arrest (*n* = 6) vs. CG (*n* = 10) are presented. Results are presented as arithmetic mean and standard deviation ($\bar{x}$ ± SD). Datasets with different asterisks differ significantly: * *p* < 0.05; ** *p* < 0.01; n.s. = not significant.

**Figure 9 ijms-24-06083-f009:**
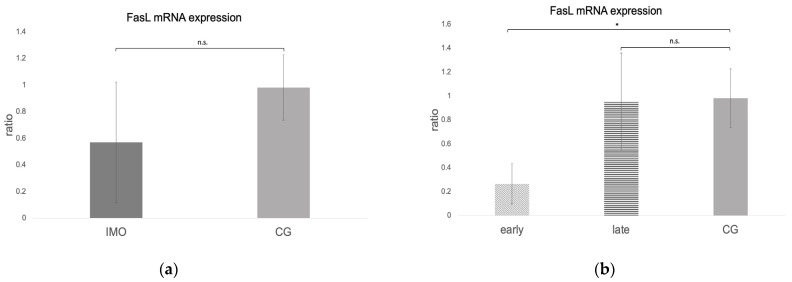
*FasL* mRNA expression (ratio) in testicular samples obtained from dogs suffering from chronic asymptomatic orchitis (CAO) and healthy control dogs (CG). (**a**) Ratios of CAO (*n* = 11) and CG (*n* = 10) and (**b**) ratios of CAO early arrest (*n* = 5) and CAO late arrest (*n* = 6) vs. CG (*n* = 10) are presented. Results are presented as arithmetic mean and standard deviation ($\bar{x}$ ± SD). Datasets with different asterisks differ significantly: * *p* < 0.05; n.s. = not significant.

**Figure 10 ijms-24-06083-f010:**
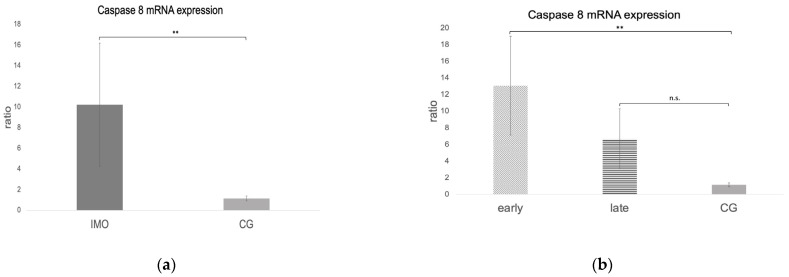
*Caspase 8* mRNA expression (ratio) in testicular samples obtained from dogs suffering from chronic asymptomatic orchitis (CAO) and healthy control dogs (CG). (**a**) Ratios of CAO (*n* = 11) and CG (*n* = 10) and (**b**) ratios of CAO early arrest (*n* = 5) and CAO late arrest (*n* = 6) vs. CG (*n* = 10) are presented. Results are presented as arithmetic mean and standard deviation ($\bar{x}$ ± SD). Datasets with different asterisks differ significantly: ** *p* < 0.01; n.s.= not significant.

**Figure 11 ijms-24-06083-f011:**
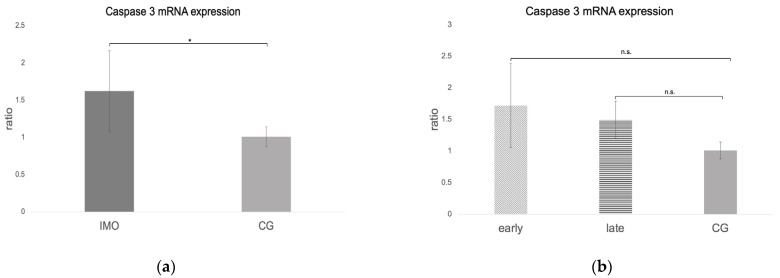
*Caspase 3* mRNA expression (ratio) in testicular samples obtained from dogs suffering from chronic asymptomatic orchitis (CAO) and healthy control dogs (CG). (**a**) Ratios of CAO (*n* = 12) and CG (*n* = 10) and (**b**) ratios of CAO early arrest (*n* = 5) and CAO late arrest (*n* = 7) vs. CG (*n* = 10) are presented. Results are presented as arithmetic mean and standard deviation ($\bar{x}$ ± SD). Datasets with different asterisks differ significantly: * *p* < 0.05; n.s.= not significant.

**Figure 12 ijms-24-06083-f012:**
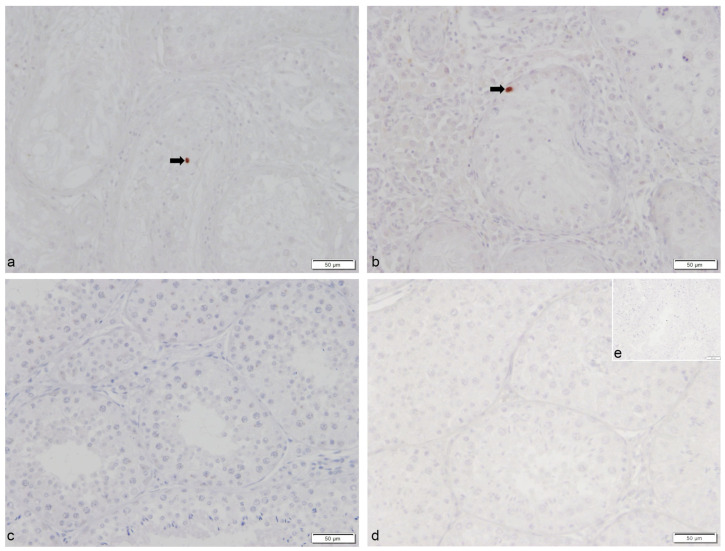
**Only sporadic caspase 3^+^ immunostaining was observed in dogs with chronic asymptomatic orchitis (CAO), whereas caspase 3^+^ staining was absent in normospermic healthy controls.** Immunostaining for cleaved *caspase 3* in (**a**,**b**) dogs with CAO (*n* = 12) and (**c**) from normospermic healthy controls (CG, *n* = 10), (**d**) isotype control, and (**e**) negative control (given as insert); ((**a**–**e**) 200× magnification) 🠮 *Caspase 3* immunopositive (caspase 3^+^) cells.

**Table 1 ijms-24-06083-t001:** Sequences of primers for RT-PCR and RT-qPCR, amplicon length, efficiency and accession number.

Primer	Accession Nr.	Forward Sequence (5′→3′)	Reverse Sequence (5′→3′)	Amplicon Length (bp)	Efficiency
*Bcl-2*	NM_ 001002949.1	ATGTGTGTGGAGAGCGTCAAC	GCCAGGAGAAGTCAAACAGAGG	175	1.98
*Bax*	NM_ 001003011.1	TCAAGCGCATCGGAGATGAAC	TCGAAGGAAGTCCAGTGTCCAG	248	1.98
*Fas*	XM_ 022410445.1	AGACCCGGAATACCAAGTGCAG	GGATGAGGACGCAAAACCACAG	180	1.91
*FasL*	NM_ 001287153.1	CAGCGAAAGGCATGTAGCACC	CACCCCAGAGACAAGGGCAAT	169	2.05
*Caspase 3*	NM_ 001003042.1	TCCAGTCACTTTGTGCGATGC	CCAAACCAAACCAAACCAACCC	218	2.02
*Caspase 8*	NM_ 001048029.1	GCTTCAGATACCAGGCAGAGC	CTCCCGGCTCAAGAGAAACTTA	115	2.19
*Caspase 9*	NM_ 001031633.1	TGTCTAGTTTGCCCACTCCCAG	TGCGAAACAGCATTAGCGACC	184	1.98
*GAPDH*	NM_001003142	GGCCAAGAGGGTCATCATCTC	GGGGCCGTCCACGGTCTTC	229	1.93
*ß-actin*	AF484115.1	GCTGTGCTGTCCCTGTATG	GCGTACCCCTCATAGATGG	98	1.92

## Data Availability

Data are available on request from the corresponding author.

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
