# Peer review of "Insights into Canine Infertility: Apoptosis in Chronic Asymptomatic Orchitis"

_ijms, 2023, doi:10.3390/ijms24076083_

Round 1

Reviewer 1 Report

The manuscript entitled “Insights into canine infertility: Apoptosis in chronic asymptomatic immune-mediated orchitis” addresses the role of apoptosis in the pathogenesis of Chronic asymptomatic immune-mediated orchitis (IMO) and associated molecular mechanisms. Initially, the authors proved that TUNEL+ cells were significantly increased in IMO versus undisturbed spermatogenesis (CG). Then, the authors proceeded to some implicated mechanisms. To this end, the authors demonstrated that the intrinsic as well as extrinsic apoptotic pathways are involved in the pathophysiology of canine IMO. Notably, Bcl-2 expression was upregulated in Sertoli cells and B-lymphocytes. The latter event is possibly linked to the resistance of B-lymphocytes to apoptosis and the perpetuation of immune cell activation in IMO.

The current findings are interesting.

 Comments:     

 1) How did the authors decide on calculating the sample size per each experimental group (12 dogs for IMO and 10 for the normal group)?

2) In section 4.1., the authors are advised to provide an ethics statement with the name of the ethical committee and ethical approval number.

3) In line 19 in the abstract, AIO is listed without describing the full name. The authors are advised to give the full name at first mention, then use an abbreviation consequently.

4) To avoid readers’ confusion in the abstract section, the authors are advised to either use only one abbreviation IMO or AIO, or to fully give the full name for both IMO and AIO and explain that asymptomatic immune-mediated orchitis (IMO) was previously called autoimmune orchitis (AIO).

5) In figures 2, 6, 7, 8, 9, 10, and 11(the gene expression data), have the authors considered that the gene expression assays of Bcl2, Bax, Fas, FasL, and caspase 3, 8, 9, using RT-PCR may not be adequate for quantifying the target signals since the mRNA expression may not necessarily reflect the corresponding protein levels due to the post-translational modifications? In fact, detecting protein signals using ELISA or Western blotting is expected to give more reliable data than gene expression assays. The authors are advised to address this limitation in the discussion section.

6) In qRT-PCR, did the authors check the RNA quality with A260/280, and perform an RT negative control to ensure no DNA contamination in the RNA extraction? Please, add these data in the relevant section (4.3) in material and methods.

7) In qRT-PCR: The authors are advised to add the gene accession number and amplicon size for all target genes. Please, add these data in the relevant section in material and methods.

8) In line 473, the authors are advised to provide table 1 as it is missing from the current manuscript.  

9) In lines 461 and 475, the authors are advised to add the cat no. for the used chemicals and kits. Please, address this issue in the entire material and methods section.

10) In TUNEL assay (section 4.5), and immunohistochemistry (section 4.6): each method should be described in detail, including image analysis (including the number of subjects, the number of images, and the software used).

11) To make all figure legends stand-alone, authors are advised to add the full name of the used abbreviations at the end of each legend. Moreover, figure captions must include information on the number of animals/replicates from which data were extracted. Authors are advised to address this point and add the answers to the relevant figure legends.

12) In figures 1, 4, 5, etc., to avoid readers’ confusion, the authors are advised to quantify the immunohistochemistry data and present the resultant data in the relevant figure with proper statistics. Without quantification, the data would be unreliable.

13) In lines 643-644 (Data availability statement), the authors state “The data presented in this study are available in Supplementary Materials of this paper”. However, no provided supplementary material in the current work. The authors are advised to double-check/modify the above statement since there is no provided “supplementary material” for the current study.

14) Careful revision of the reference list should be performed. For example, reference no. 85 lacks page numbers.

15) Again, the authors are advised to carefully revise the reference section. The authors are advised to unify the way they write the journal name. Sometimes it is written as an abbreviation (reference 86) while in other references (such as reference 85) it was written as a full name.

Author Response

Dear reviewer!

Thank you very much for your careful and positive review of our work. Please find our detailed answer attached 

Sincerely,

Sandra Goericke-Pesch

NB As I am not 100% sure if the letter is the latest version and I could not change it, I attach my answer also here

Dear reviewer 1,

thank you very much for the careful and positive review of our manuscript. This is very much appreciated! We answered all your comments and questions in detail below.

We sincerely hope that with the respective changes, you can support publication of our manuscript.

Looking forward to your response and thank you again for your efforts.

Sincerely,

1) How did the authors decide on calculating the sample size per each experimental group (12 dogs for IMO and 10 for the normal group)? 

The number of animals is (unfortunately) not based on statistical calculation but based on availability of affected dogs. As our IMO group was not experimentally induced but occurring spontaneously, we decided to include 12 canine samples that we collected for diagnostic purposes that followed the inclusion criteria for chronic asymptomatic immune-mediated orchitis. Immune-mediated because no causative condition or agent could be identified at time of presentation and chronic asymptomatic because dogs according to the owner have never showed clinical signs of disease, pain, fever etc. and no clinical abnormality except for softened and smaller testes was identified at time of presentation. To date, no method of induction of chronic asymptomatic (immune-mediated) orchitis is known in the dog. To have a comparable number of healthy controls with physiological spermatogenesis, we decided to include 10 dogs in the normal group. We are well aware that larger group sizes would have been even better and would have given our data more power, but we believe that our group sizes and study design are suitable to obtain (initial preliminary) valuable insights. We hope that you agree and added the group size as a limitation to the manuscript.

2) In section 4.1., the authors are advised to provide an ethics statement with the name of the ethical committee and ethical approval number.

Permission for all samples was received by the respective authorities. Due to collection of IMO samples for diagnostic purposes, no animal experimentation approval was required and so was not for elective surgical castration of healthy control animals (as it was the owners’ choice and not related to the investigation). All required details had been added to the manuscript now. We hope you are satisfied.

3) In line 19 in the abstract, AIO is listed without describing the full name. The authors are advised to give the full name at first mention, then use an abbreviation consequently.

I apologize for this, AIO was replaced by IMO. After the comments of another reviewer who we can for sure say that it is immune mediated, we changed the abbreviation to chronic asymptomatic orchitis (CAO).

4) To avoid readers’ confusion in the abstract section, the authors are advised to either use only one abbreviation IMO or AIO, or to fully give the full name for both IMO and AIO and explain that asymptomatic immune-mediated orchitis (IMO) was previously called autoimmune orchitis (AIO).

I agree and it was a mistake that both terms were mentioned in the abstract. Every AIO is now replaced by IMO. As explained above this was changed to CAO.

5) In figures 2, 6, 7, 8, 9, 10, and 11(the gene expression data), have the authors considered that the gene expression assays of Bcl2, Bax, Fas, FasL, and caspase 3, 8, 9, using RT-PCR may not be adequate for quantifying the target signals since the mRNA expression may not necessarily reflect the corresponding protein levels due to the post-translational modifications? In fact, detecting protein signals using ELISA or Western blotting is expected to give more reliable data than gene expression assays. The authors are advised to address this limitation in the discussion section. 

I completely agree to you comment. Unfortunately, despite trying various antibodies for the different targets, we failed to identify cross-reacting antibodies for most targets. Besides the amount of tissue taken at the biopsy collection was too small. Due to this, Western Blot analysis and protein quantification was not possible and IHC was only performed for few targets. Custom-made antibodies might be considered in future studies, not only for IHC, but also to quantify protein expression using Western Blot. We had a passage about this in the discussion, but we specified this aspect further according to your suggestion. 

6) In qRT-PCR, did the authors check the RNA quality with A260/280, and perform an RT negative control to ensure no DNA contamination in the RNA extraction? Please, add these data in the relevant section (4.3) in material and methods.

 This was done and is added accordingly.

7) In qRT-PCR: The authors are advised to add the gene accession number and amplicon size for all target genes. Please, add these data in the relevant section in material and methods.

I sincerely apologize that table 1 that we referred to in the text was not included in the manuscript. This is now added. It contains all relevant information. We hope you are satisfied.

8) In line 473, the authors are advised to provide table 1 as it is missing from the current manuscript.  

Please see above, this table contains all information, unfortunately we forgot to transfer it to the IJMS template.

9) In lines 461 and 475, the authors are advised to add the cat no. for the used chemicals and kits. Please, address this issue in the entire material and methods section. 

 All relevant catalogue numbers were added now.

10) In TUNEL assay (section 4.5), and immunohistochemistry (section 4.6): each method should be described in detail, including image analysis (including the number of subjects, the number of images, and the software used). 

 This information is given in detail in the Materials and Method section: For TUNEL, Bcl2 and Caspase3 (all IHCs performed) sections of all animals (IMO: n=12, CG: n=10) were included and evaluated. If number of animals differed for a parameter, it was mentioned in the result section. However, this was only the case for some genes where one dog had to be excluded (because of negative results), not for IHC. For descriptive evaluation, slides were completely checked for staining etc. In case the immunopositive signal was quantified, number of visual fields (including magnification) and/or tubules evaluated are given in the respective section in the Materials and Method part. Additionally to the microscope used, the camera and the software was now added.

11) To make all figure legends stand-alone, authors are advised to add the full name of the used abbreviations at the end of each legend. Moreover, figure captions must include information on the number of animals/replicates from which data were extracted. Authors are advised to address this point and add the answers to the relevant figure legends.

 We have modified the figure legends; all abbreviations are explained and number of animals was added to all figures.

12) In figures 1, 4, 5, etc., to avoid readers’ confusion, the authors are advised to quantify the immunohistochemistry data and present the resultant data in the relevant figure with proper statistics. Without quantification, the data would be unreliable.

For all parameters where IHC staining was quantified, statistical analysis is given in the result section. We added the graph about number of TUNEL+ cells in visual fields to figure 1 (f, g) as we referred to this in the text earlier, too, and we agree that this addition facilitates the understanding of data for the reader.

13) In lines 643-644 (Data availability statement), the authors state “The data presented in this study are available in Supplementary Materials of this paper”. However, no provided supplementary material in the current work. The authors are advised to double-check/modify the above statement since there is no provided “supplementary material” for the current study. 

I apologize for this. It was modified to “Data are available on request from the corresponding author.”

14) Careful revision of the reference list should be performed. For example, reference no. 85 lacks page numbers. 15) Again, the authors are advised to carefully revise the reference section. The authors are advised to unify the way they write the journal name. Sometimes it is written as an abbreviation (reference 86) while in other references (such as reference 85) it was written as a full name

I revised the reference list carefully and corrected it. I hope you are satisfied and apologize for this.

Reviewer 2 Report

in the submitted manuscript, S. Goericke-Pesch and her team present new information about a possible relationship between apoptosis (both intrinsic and extrinsic pathways) and immuno-mediated orchitis in dogs.

This paper has the potential to provide innovative information, but I have several concerns on regard to the content of the paper.

The major one, that needs the authors' full attention, respects the characterization of the condition "asymptomatic immune-mediated orchitis" in dogs. how was it diagnosed? I could not find enough information in M&M section that would allow distinguishing between IMO and testicular atresia or any other form of chronic orchitis in dogs.

This is an important issue, as apoptosis is increased both in tubular atresia and chronic orchitis, the later also increasing the number of inflammatory cells in the testicular interstitium (represented by B-cells in some conditions).

Additional concerns respect the fact that in some pictures (e.g., fig 1 and fig 4, existing background exists, compromising the ability to interpret the results of the immunohistochemistry (considering that positivity was scored, among others, as fair, matching that kind of reaction). In  figure 5,  Sertoli cells are identified as (intratubular) B-cells but the nuclear morphology identifies the cell as a Sertoli. In these images,  both Sertoli cells and spermatogonia show a faint positive labeling for CD20, suggesting that either the antibody is producing some cross-labeling, or the technique somehow was properly run. Authors should recheck their data for CD20 labeling or maybe confirm the results by using a different primary antibody. besides CD20 antibody may label a fragment of CD20 (the L26 fragment) that has been reported outside B-cells.

A question remains: why the authors did not include the protein content for several molecules used in the study? that would allow assessing the post-transcriptional regulation of the targeted genes.

In material and methods, the authors should provide additional information (besides the information demanded above) respecting:

- For IHC, the authors should 1) identify the positive controls used for each primary antibody (not only the negative). Particularly for molecules whose expression in the slides is sporadic; 2) identify the positive controls used for each primary antibody (not only the negative). In particular for molecules whose expression in the slides is sporadic; 3) provide the complete reference for the primary antibodies used as well as the conditions for the reaction (both time and temperature of incubation)

- IHC - the authors describe the use of the Vector Nova-RED Substrate Kit SK-4800 for revelation. this kit would stain in red the immune reaction against the targeted molecule in IHC. However, in the pictures provided, the immunoreaction and the background is brown. Please, recheck if DAb was used instead of the Nova-Red.

- Jurkat cells (or humane origin) are not a proper positive control for canine tissues WB. Authors should provide information regarding the validation of these cells for controls in canine tissues, or present a valid positive control

 Finally, the use of different acronyms for the same condition makes it difficult to understand the message. please select one and stick with it

Additional minor comments can be found in the attached copy of the MS

Author Response

Dear reviewer,

thank you very much for your valuable comments. Please find attached our comments point by point in the separate letter.

We hope you are satisfied with the updated version and can agree on publication.

Sincerely,

Sandra Goericke-Pesch

Reviewer 3 Report

Manuscript entitled :Insight into canine infertility: Apoptosis in chronic asymptomatic immune-mediated orchitis by Morawietz J et all., which is taken into consideration to publish in IJMC contain valuable data pertaining to the mechanism of IMO in dogs. It is another step in the search for a mechanism of infertility in dogs as well as the prove that the dog’s IMO/AIO may be suitable model to investigate the same problems in human. I have only minor concerns listed below:

1. Line 78-79 Please add information that T cells were previously examined in the pathogenesis of IMO in the Author’s study and add the rationale for tracking B cells involvement in IMO

2. Figure’s captures: please add description of the stage of spermatogenic arrest whenever it is possible. In some figures it is clearly seen that (a) sample belongs to early stage of spermatogenesis arrest, but it is not written.

3. Figure 4: CG section show a small portion of interstitial area. In my opinion, it would be more informative if these sections reveal more of that tissue to prove the lack of Bcl-2 staining in interstitial compartment

4. Line 389-391: Please explain potential mechanisms of lymphocyte infiltration into tubular compartment and the cause of blood-testis-barrier breake-down in IMO

5. Line 473: There is a reference to table 1, but no table in manuscript

6. References: (15)- please complete this reference; (92)- please consider to omit conference abstracts in the literature cited; (100) – I think it is better to use reference 7, as it was previously used in the description of tissue processing. Especially that in that paper testicular tissue was used instead of uterine ones. Please avoid to cite Your own papers only to gain more citations.  

Author Response

Dear reviewer,

thank you very much for your thorough review. Please find attached our point-by-point answer attached. 
We hope that you consider the manuscript suitable for publication

Best

Sandra

Round 2

Reviewer 2 Report

In the revised MS, some concerns have been addressed by the authors. However, in the introduction, the authors still insist on pressing a putative immune-mediated "label" into the clinical condition. Without proving the existence of auto-antibodies, the immune-mediated etiology of the process can not be established. Since no reference is made in all the team MS on the topic of asymptomatic chronic orchitis in dogs to the detection of said antibodies, all the authors have is a diagnosis of chronic, asymptomatic orchitis. Therefore, this section must be revised accordingly. If I am mistaken, and indeed, the existence of those antibodies was searched, I apologize for the conclusions.

Another major concern of mine is the fact that multiple images of TUNEL assay and Immunohistochemistry present a background. The intensity of the background in some of them is equivalent to a low level of positivity ascribed to the slide in another case. The IHC (and Tunel also) is a technique difficult to standardize, particularly when using polyclonal primary antibodies. Nonetheless, an effort must be made to avoid possible biases regarding the existence of unspecific labeling or remainder endogenous peroxidase activity (frequent causes for the background). For that, quality control is critical.  As it is I would like to propose the authors confirm their results for the slides presenting background and produce new, clear images. As they are now, I am not confident with the results.

Also, It is important to acknowledge that the TUNEL assay is not "full proof" to mark apoptosis, as the process can be reverted in Tunel-positive cells.

Additional comments were introduced in the commented file attached to this review

Author Response

Dear reviewer,

please find attached the cover letter with the revision notes and the highlighted and non-highlighted version of the manuscript. Please note that the abstract in the document is changed, but cannot be changed above.

Sincerely, Sandra Goericke-Pesch

Round 3

Reviewer 2 Report

In the revised MS now submitted the authors improved the text. However, they still put the focus of the introduction and discussion in autoimmune orchitis.

As I understand, the presence of autoantibodies was not tested, and therefore there is no means to prove the samples used in this study are representative of an autoimmune condition. The authors may have a hint or a feeling, but if they can not prove that autoimmune was at the origin of the process. They can not assume or hypothesize. Since the tissue morphology is similar between autoimmune orchitis (AIO) and chronic asymptomatic orchitis (CAO), the authors should not narrow the introduction and discussion sections in AIO. Besides, apoptosis is also increased in CAO. Figure 4 a & b show degenerative changes of the seminiferous tubules, which are frequently found in chronic non-auto-immune orchitis.

Therefore, I strongly recommend the authors to revise the introduction and discussion section to include information regarding apoptosis in CAO, and  avoid putting the focus on AIO:

Another suggestion is to move the text and images regarding the WB from the Results section into the material and methods, since it was meaning to validate the use of the antibody

Author Response

Please find our response letter attached.

Thank you for your efforts!

Sincerely 

Sandra Goericke.Pesch
